# The Human Cytomegalovirus β2.7 Long Non-Coding RNA Prevents Induction of Reactive Oxygen Species to Maintain Viral Gene Silencing during Latency

**DOI:** 10.3390/ijms231911017

**Published:** 2022-09-20

**Authors:** Marianne R. Perera, John H. Sinclair

**Affiliations:** Cambridge Institute of Therapeutic Immunology and Infectious Disease, Department of Medicine, University of Cambridge, Addenbrooke’s Hospital, Hills Road, Cambridge CB2 0QQ, UK

**Keywords:** HCMV, latency, non-coding RNA, reactive oxygen species, β2.7

## Abstract

Human cytomegalovirus (HCMV) is a significant source of disease for the immunosuppressed and immunonaive. The treatment of HCMV is made more problematic by viral latency, a lifecycle stage in which the virus reduces its own gene expression and produces no infectious virus. The most highly expressed viral gene during HCMV latency is the viral β2.7 long non-coding RNA. Although we have recently shown that the β2.7 lncRNA lowers levels of reactive oxygen species (ROS) during infection in monocytes, how this impacts latency is unclear. We now show that β2.7 is important for establishing and maintaining HCMV latency by aiding the suppression of viral lytic gene expression and that this is directly related to its ability to quench reactive oxygen species (ROS). Consistent with this, we also find that exogenous inducers of ROS cause reactivation of latent HCMV. These effects can be compensated by treatment with an antioxidant to lower ROS levels. Finally, we show that ROS-mediated reactivation is independent of myeloid differentiation, but instead relies on NF-κB activation. Altogether, these results reveal a novel factor that is central to the complex process that underpins HCMV latency. These findings may be of particular relevance in the transplant setting, in which transplanted tissue/organs are subject to very high ROS levels, and HCMV reactivation poses a significant threat.

## 1. Introduction

Human cytomegalovirus (HCMV) is a common herpesvirus that infects 50–100% of the population [1]. Despite causing only asymptomatic/mild illness in the majority of people, the virus is a source of significant disease for the immunosuppressed (including transplant recipients) and the immunonaive foetus [2]. HCMV has two lifecycles: lytic and latent. In a lytic infection, the virus expresses a cascade of temporally regulated genes: immediate early, early and late, and produces infectious virus. In contrast, in cells of the early myeloid lineage, such as CD34+ haematopoietic stem cells and CD14+ monocytes, the lytic replication cycle is suppressed and, instead, the virus establishes a latent infection [3,4]. Latent virus produces no infectious progeny and shows greatly reduced levels of viral gene expression, allowing it to evade immune detection [5]. This reduction in HCMV lytic gene expression during latency is mediated by silencing of the viral major immediate early promoter (MIEP), which suppresses the expression of major immediate early proteins, IE1 and IE2, preventing them from transactivating other lytic viral gene promoters [6]. Virus reactivation from latency that involves MIEP is de-repression, which allows lytic infection to resume [7]. Cues to this reactivation are known to include differentiation of the host cell to a macrophage or dendritic cell as well as inflammation [8,9,10].

Although latent infection is generally associated with suppression of viral major immediate early (IE) gene expression, a number of viral genes are now known to be expressed during latency, though at much lower levels than during lytic infection [11]. The most abundant of these is a viral long non-coding RNA called β2.7. Long non-coding RNAs are RNAs over 200 nucleotides long with little to no protein coding potential [11,12]. In this paper, we have examined the role of β2.7 in viral gene silencing during latency. We have previously shown that the β2.7 lncRNA reduces levels of reactive oxygen species (ROS) during infection in monocytes [13]. Here, we show that this suppression of ROS production by β2.7 is essential to establish and maintain HCMV latency. We also find that exogenous stimulators of ROS can trigger reactivation of latent HCMV. It is possible that HCMV evolved to reactivate in response to ROS as ROS levels are high in tissue locations that are of particular biological relevance for virus reactivation. Additionally, this could represent a “quick exit” strategy for latent virus to reactivate in the face of oxidative stress and cell death. Altogether, these results reveal a novel mechanism that is central to the complex process that underpins HCMV latency. This may be important in the transplant setting, in which transplanted tissue/organs experience very high ROS levels [14], and HCMV reactivation can cause significant disease [15,16].

## 2. Results

### 2.1. The Absence of β2.7 Results in Increased GFP Expression in a Sub-Population of Infected CD14+ Monocytes

Recombinant HCMV carrying a GFP reporter under the control of the SV40 promoter (WT SV40-GFP TB40E) has been used by many to identify latently infected myeloid cells in experimental models of latency [17]. Upon infection of, e.g., monocytes with WT SV40-GFP TB40E, GFP-positive cells are routinely observed in the absence of production of infectious virions (see Figure 1A)—a classical hallmark of latency. This GFP expression is gradually silenced alongside the rest of the viral genome during latency [18,19]. However, on infection of monocytes with an SV40-GFP tagged TB40E virus deleted for β2.7 (Δβ2.7 SV40-GFP TB40E) we noted that a substantial proportion of infected cells expressed GFP to much higher levels (see Figure 1A) and failed to silence their GFP over time (Figure 1B).

Whilst the difference in levels of GFP expression between WT SV40-GFP TB40E and its Δβ2.7 deleted counterpart was clear in infected monocytes, no such differences were observed in lytically infected fibroblasts (Figure 1C). This suggests that GFP expression by the deletion virus was not inherently defective, but this phenomenon was specific to viral infection in monocytic cells. 

Interestingly, higher levels of GFP expression from WT SV40-GFP TB40E virus are often seen in infected differentiated monocytes undergoing a lytic, rather than latent, infection as the viral genome is both de-repressed and amplified (see Figure 1D). Therefore, we hypothesized that the absence of β2.7 might impair the ability of the virus to silence its genome during latency.

### 2.2. Viral Major Immediate Early Gene Expression Is Not Silenced in Δβ2.7 Infected Monocytes

To probe this further, we tested whether the major immediate early genes were still being silenced in monocytes infected with Δβ2.7 virus. We infected CD14+ monocytes with either the TB40E SV40-GFP WT or Δβ2.7 viruses. At 7 or 12 d.p.i, we harvested RNA from these cells and used RTqPCR to assay them for levels of the viral major IE genes IE72/IE86, which are under control of the MIEP. As shown in Figure 2A, we found a significant increase in the levels of IE RNA present in Δβ2.7-infected monocytes relative to WT, suggesting that there was indeed a failure to silence the MIEP in the absence of β2.7 upon infection of primary monocytes. 

To ensure that we had used an equivalent amount of WT and Δβ2.7 virus in these experiments, we also measured levels of viral genome within the monocytes by harvesting DNA from infected cells at 1 d.p.i. Cells were first citrate washed to remove any bound extracellular virions that had failed to enter the monocytes (as described in [20]) and then viral genomes were quantified by qPCR targeting the promoter regions of the viral gene UL44 and the housekeeping gene, GAPDH. Reassuringly, we found no significant difference in genome levels between WT and Δβ2.7 viruses in the infected monocytes at 1 d.p.i (see Figure 2B), suggesting that the differences in IE levels shown in Figure 2A were not simply a result of more input Δβ2.7 virus.

In addition to measuring total IE RNA levels, we tested whether the number of monocytes that stained positive for IE protein dropped over time as occurs during a WT infection when the viral genome is silenced following an initial burst of expression [21,22]. We infected CD14+ monocytes with either TB40E SV40-GFP WT or Δβ2.7 virus and fixed cells at 3, 6 or 9 d.p.i, stained cells for IE1/2 protein, and counted the number of IE-positive cells per well. As shown in Figure 2C, we observed a much more modest drop in the number of IE-positive cells in Δβ2.7 infection compared to WT, again suggesting that the ability to silence the MIEP in monocytes was impaired in the absence of β2.7. Although the difference between WT and Δβ2.7 becomes less obvious at 9 d.p.i, we speculate that some of the decline in IE-positive or GFP-positive cells may be attributable to cell death, which is known to happen at higher rates in Δβ2.7 infected cells under stress [13,23].

### 2.3. Δβ2.7 Infection Results in a Lytic Rather than Latent Infection in a Portion of Infected Monocytes

It is known that IE expression is not always predictive of a full lytic replication programme and can result in abortive infection in which IE gene expression can be detected but with little subsequent early/late gene expression [24,25]. Consequently, we investigated how far the virus was progressing through its lytic replication cycle in the absence of β2.7 and stained WT or Δβ2.7 infected monocytes for pp28, a viral protein that is only expressed in the late phase of lytic infection. Whilst there was little pp28 in WT virus-infected monocytes, we saw significantly more pp28-positive monocytes in Δβ2.7 infection (Figure 3A,B). We also repeated this analysis using the Toledo strain of HCMV, and again found that there were significantly more pp28-positive cells in Δβ2.7-infected monocytes (see Figure 3C). 

We then tested whether monocytes infected with Δβ2.7 virus were making infectious virions. CD14+ monocytes were infected with GFP-tagged TB40E WT or Δβ2.7 deletion virus and then cocultured with indicator HFFs, which are permissive for a full lytic infection (see Figure 4A for schematic). 

Figure 4B (upper left panel) shows a single green, fluorescent monocyte infected with WT virus that fails to produce a plaque in the surrounding fibroblasts, which is consistent with it being a latently infected cell. However, monocytes infected with Δβ2.7 virus were routinely surrounded by plaques, indicating the release of infectious virions (Figure 4B, upper right panel). Quantification of these analyses shows significantly more GFP-positive infected plaques arising from Δβ2.7 infected monocytes (see Figure 4C). Our findings were the same when the analysis was repeated with the Toledo strain of HCMV, but plaque formation was identified only by pp28 staining as the Toledo virus was not GFP-tagged (Figure 4B, lower panel, and Figure 4D). Altogether, these results show that β2.7 promotes the silencing of lytic viral genes during HCMV latency in CD14+ monocytes, and in its absence, a portion of monocytes undergo lytic rather than latent HCMV infection. 

### 2.4. A Reactive Oxygen Species Inducer, Rotenone, Prevents HCMV Latency Establishment in a Sub-Population of Infected CD14+ Monocytes

Given that we have previously shown that β2.7 reduces levels of reactive oxygen species in monocytes [13], we first tested whether the failure to maintain latency in a Δβ2.7 infection might be related to higher levels of intracellular ROS. ROS are known to affect many signalling pathways and they are also involved in inflammation and myeloid cell differentiation, which are both coupled to reactivation of HCMV [9,10,26,27]. 

Therefore, we tested whether artificially increasing ROS levels in WT infection would lead to a lytic infection in HCMV-infected monocytes. Using a dye for actively respiring mitochondria (TMRE), we first determined that the mitochondrial complex I inhibitor, rotenone (a widely used ROS inducer [28,29,30]), caused minimal cell death at lower doses (Appendix A) whilst still inducing detectable ROS production in monocytes (Figure 5A). As expected, when these cells were co-treated with the antioxidant, N-acetyl cysteine (NAC), the ROS levels decreased (Figure 5A). 

We then infected monocytes with TB40E SV40-GFP WT HCMV, and at 3 h.p.i, either left them untreated, treated them with two different concentrations of rotenone, or treated the cells with IL-4 and GM-CSF; IL-4/GM-CSF which differentiates the cells and acts as a positive control (since differentiated monocytes support a lytic infection). 7 d.p.i, cells were stained for the lytic protein, pp28. After treatment with rotenone, significantly more infected monocytes expressed pp28 compared to untreated cells, suggesting rotenone treatment of monocytes encourages the initiation of a lytic rather than latent infection (see Figure 5B).

### 2.5. Reactive Oxygen Species Inducer, Rotenone, Reactivates Latent HCMV in CD14+ Monocytes

Clearly, our results so far showed that rotenone treatment at the time of infection could prevent HCMV from establishing latent infection. However, we also wanted to determine whether rotenone would reactivate virus that had already established latency. Consequently, we repeated the previous experiment but added rotenone at 0 (3 h.p.i), 2 and 4 d.p.i and then assayed for late pp28 protein expression at 7 d.p.i. This showed a significant increase in the number of infected monocytes expressing late lytic pp28 at all timepoints of rotenone treatment compared to untreated latently infected cells (Figure 6A). This argues that that rotenone is not only capable of preventing the establishment of latency, but can also reactivate virus from latency once it has been established.

### 2.6. An Antioxidant Curbs Rotenone-Induced Lytic Gene Expression in Otherwise Latently Infected Monocytes

Rotenone can have effects other than increasing ROS levels, for example it can stop ATP production by halting the electron transport chain [31]. We therefore ensured that HCMV reactivation was a consequence of ROS specifically, and not other effects of rotenone. On the basis that that the antioxidant NAC reduces superoxide levels incurred by rotenone treatment (Figure 5A), we tested whether NAC could suppress the induction of lytic gene expression in rotenone treated WT virus-infected monocytes. Figure 6B clearly shows that NAC decreases levels of rotenone-induced virus reactivation, lending support to the idea that it is ROS that are driving reactivation events.

### 2.7. Direct Sources of ROS Can Also Prevent the Establishment/Maintenance of Latency in Primary Monocytes

We next wanted to confirm that multiple sources of ROS would also reactivate HCMV in latently infected monocytes. Consequently, we treated WT virus infected monocytes with increasing concentrations of TBHP (tert-Butyl hydroperoxide), which is a cell permeable analogue of hydrogen peroxide and, itself, a form of ROS [32]. At 7 d.p.i, we quantified expression of the late lytic protein pp28 by immunofluorescent staining and observed that TBHP significantly increased the levels of lytic infection in WT infected monocytes (Figure 7).

### 2.8. An Antioxidant Reduces the Amount of Lytic Infection in Δβ2.7 Infection in Primary Monocytes

To confirm that Δβ2.7 failed to establish/maintain a latent infection due to high ROS levels, we then tested whether NAC treatment would rescue viral gene silencing in Δβ2.7 virus-infected monocytes. As shown in Figure 8, NAC did indeed compensate for the β2.7 deletion and significantly reduced the number of Δβ2.7 virus infected cells that failed to undergo a latent infection. To ensure NAC had no effect on viral entry, we also confirmed that we saw equivalent levels of viral genome in cells pretreated with and without NAC (see Appendix A). Taken together, the results presented so far suggest that ROS have an important role in regulating lytic viral gene expression during the establishment and maintenance of latency as cells with high levels of ROS, mediated by external stimuli, do not fully repress the viral MIEP and fail to support latent infection. Additionally, the presence of the viral β2.7 gene helps suppress this ROS-mediated MIEP activation and helps to prevent lytic productive infection.

### 2.9. Higher ROS Levels Do Not Induce Differentiation of the Infected Monocytes

Differentiation of CD34+ progenitors or monocytes into macrophages or dendritic cells is known to result in reactivation of latent HCMV (reviewed in [26]). Of note, ROS have been reported to be essential for the differentiation of monocytes into DCs and macrophages [33], both of which are able to support a lytic HCMV infection [34,35,36]. Inducers of monocyte differentiation, such as M-CSF and GM-CSF, or inducers of monocyte maturation, such as LPS and TNFα, are all known to robustly stimulate ROS production [34,37,38,39,40]. Indeed, ROS induced by GM-CSF or M-CSF are thought to have a critical role in how these cytokines mediate monocyte differentiation to M2 macrophages [38,41]. We also show in Appendix A that another widely used inducer of monocytic cell differentiation, PMA, also causes ROS production; these ROS have also been shown to be essential in the differentiation of monocytic THP-1 cells [42]. 

Consequently, we asked whether Δβ2.7-infected cells or rotenone-treated, WT virus-infected cells were reactivating because high levels of ROS were simply resulting in monocyte differentiation. To investigate this, we analysed Δβ2.7-infected or rotenone-treated monocytes for differentiation markers. 

Untreated CD14+ monocytes, monocytes differentiated with IL-4/GM-CSF and matured with LPS or monocytes treated with two different concentrations of rotenone for 5 or 7 days (to mimic conditions from the reactivation experiment shown in Figure 5 and Figure 6) were stained for 4 different myeloid differentiation markers: CD86 (upregulated on the surface of M1 macrophages and monocyte-derived dendritic cells [43,44,45]), DC-SIGN (upregulated on monocyte-derived dendritic cells [46]), CD206 (upregulated on M2 macrophages [47]) and HLA-DR (upregulated on M1 macrophages and mature dendritic cells [48,49]). As shown in Figure 9, we observed no rotenone-induced increase in these differentiation markers, suggesting it was unlikely that the rotenone was simply differentiating the monocytes to virus permit reactivation. 

We also stained monocytes infected with TB40E SV40-GFP WT or Δβ2.7 virus for the same differentiation markers by immunofluorescence. Figure 10 shows that there was no evidence of DC-SIGN, MMR or CD86 expression in any mock, WT or Δβ2.7 infected monocytes, despite them staining clearly as expected in the differentiated positive controls. We observed low levels of HLA-DR on undifferentiated monocytes, and this increased somewhat on differentiated cells (Figure 10). Interestingly, cells that were infected (GFP-positive) showed lower HLA-DR levels with both WT and Δβ2.7 virus infection. This was not unexpected as both latent and lytic HCMV are known to downregulate HLA-DR to avoid immune recognition [50,51,52]. Overall, our results suggested that Δβ2.7 virus-infected monocytes or rotenone treated, WT virus-infected monocytes did not fail to undergo latent infection simply because host monocytes had differentiated.

### 2.10. Δβ2.7 Infection and Rotenone Activate NF-κB

We then hypothesized that higher ROS levels could prevent latent infection by affecting important signalling pathways already known to be involved in control of latency and reactivation of HCMV in myeloid cells. ROS themselves are known to be cell signalling molecules that can have important and pleiotropic effects, including activating NF-κB [53], ERK1/2 [54], JNK [54], p38 MAPK [54], some of which have been reported to feed into MIEP activation during reactivation from latency [8,10,27,52]. 

To investigate whether any of these signalling pathways were involved in ROS-mediated reactivation of HCMV, we treated Δβ2.7-infected monocytes with different concentrations of inhibitors of ERK signalling (MEK1+2 inhibitor), p38 MAPK signalling, c-fos and NF-κB signalling, and quantified levels of pp28 in these cells (Figure 11A). We observed no discernible change to the number of pp28-positive monocytes using ERK (Figure 11A(i)), p38 MAPK (Figure 11A(ii)) pathway or c-fos inhibitors (Figure 11A(iii)). However, the NF-κB inhibitor (Figure 11A(iv)) caused a clear concentration-dependent decrease in the number of pp28-positive Δβ2.7-infected monocytes.

Consistent with a role for NF-κB in this rotenone-induced reactivation of viral IE gene expression, we analysed NF-κB localisation in Δβ2.7 virus infected monocytes or WT virus-infected monocytes treated with rotenone. Figure 11B shows that Δβ2.7 virus infection or rotenone treatment of WT-infected cells resulted in nuclear migration of NF-κB. Altogether, our data suggests that high ROS levels caused by a lack of the lncRNA β2.7, or by exogenous ROS inducers, result in activation of NF-κB which prevents establishment of latency or mediates reactivation of latently infected CD14+ monocytes.

## 3. Discussion

Understanding the factors involved in establishing and maintaining the latent state of HCMV is crucial for our full understanding of this important human pathogen. In this paper, we have examined the role of a viral long non-coding RNA, β2.7, which is the most highly expressed viral gene during HCMV latency, and have identified that this viral gene product has a primary role in establishing and maintaining viral latency in myeloid cells by hijacking the cellular ROS response.

Firstly, we show that HCMV lacking the viral β2.7 gene has a reduced ability to establish latency in primary monocytes because of an inability of the virus to suppress high levels of ROS production upon infection. Consistent with this, we observe that treating virus-infected monocytes with exogenous sources of ROS prevents the virus from suppressing viral MIEP activity resulting in a full lytic infection. Similarly, we observed that ROS induction in monocytes already latently infected with HCMV, resulted in untimely reactivation of virus from an already established latent infection. Finally, treatment of monocytes with antioxidants prevents ROS inducers from driving a lytic infection, or reactivation from latency, in latently infected monocytes.

It is well established that whilst high levels of ROS can cause irreparable cell damage (oxidative stress), at low levels, ROS are important cellular signalling molecules. Our observations are consistent with the view that HCMV has hijacked the ROS signalling pathway to make it a key player in the establishment and maintenance of latency as well as a potent signal for reactivation, potentially independent of differentiation.

Consistent with this view, it is essential that HCMV reactivates from its sites of latency in vivo, and this is known to occur in terminally differentiated myeloid cells which often occurs when monocytes travel to peripheral tissue, resulting in their differentiation into macrophages or dendritic cells [26,55]. This differentiation process elicits high intracellular levels of ROS in monocytes [34,36,41,56]. Furthermore, the sites of monocyte extravasation are usually inflammatory environments [57], which also contain high levels of ROS, generated by proinflammatory cytokines such as TNFα, the respiratory burst and cytokines that mediate monocytic differentiation. Therefore, as monocytes enter peripheral tissues, high ROS levels could act as an important signal to latent HCMV that the infected cell has disseminated to advantageous biological sites for reactivation. Conversely, and importantly, it should be noted that ROS levels are low in the bone marrow, which remains as a reservoir for latent HCMV [58]. 

A second reason that HCMV may have evolved to reactivate from latency in monocytes in response to ROS, is that if ROS levels are high (e.g., under mitochondrial stress, during differentiation or inflammation or from innate immune defences) and the latent monocyte is at potential risk of cell death, it would be beneficial for the virus to reactivate, replicate and disseminate. Such induction of lytic infection would also result in the expression of a number of anti-apoptotic genes which are known to be expressed during lytic infection but have not been detected by current in vitro models of latency [11]: for example, UL37x1 [59,60] and UL36 [61]. 

The gammaherpesviruses have also been shown to reactivate in environments with higher ROS levels. The addition of hydrogen peroxide to cells latently infected with Kaposi’s Sarcoma Herpesvirus (KSHV) activates JNK and p38 MAPKs and results in viral reactivation [62,63]. The inhibition of the transcription factor FoxO1, which upregulates the antioxidant enzyme SOD2, leads to an increase in intracellular ROS levels and reactivation of KSHV [64]. Epstein Bar Virus (EBV) also reactivates under conditions of oxidative stress via activation of JNK and p38 MAPKs [65,66]. Whilst the different subfamilies of Herpesviridae have diverged considerably, they still maintain many similarities in their life cycles, and the work presented here suggests that, like the gammaherpesviruses, ROS also play a key role in reactivation of latent HCMV. 

Interestingly, ROS have also been shown to be useful in the initiation of lytic HCMV infection in permissive cells. Binding of the HCMV virion to smooth muscle cells induces ROS within minutes, and ROS levels remain high for at least 2 h post infection [67,68]. Increased ROS levels in lytically infected cells results in rapid activation of NF-κB which subsequently binds and activates the MIEP during lytic infection [68]. This is entirely consisted with our observations that shows that ROS-mediated reactivation of latently infected monocytes results from NF-κB activation. 

It is interesting that inflammation and differentiation, which both reactivate latent HCMV, also involve high levels of ROS. Differentiation inducers such as GM-CSF, M-CSF and LPS cause increased ROS levels [37,53,69], as do proinflammatory cytokines such as TNFα. Indeed, we demonstrated here in Appendix A, that PMA, which is commonly used to differentiate myeloid THP-1 and Kasumi-3 cells, results in superoxide production in monocytes, which is in agreement with reports from other groups [70,71,72]. One question which arises from this observation is: do ROS contribute to reactivation of HCMV by differentiation/proinflammatory cytokines? However, as ROS are implicated in the monocyte differentiation process itself [33], this may prove challenging to untangle. It is also important to note that aside from ROS, there are many other mechanisms of HCMV reactivation (reviewed in [73]). For example, MAPK signalling is known to activate the transcription factor AP-1, which subsequently activates the HCMV MIEP [74]. Additionally, Src family kinases have been shown to recruit a histone acetyltransferase to the MIEP which leads to its de-repression [75]. 

One final point of interest is that ischaemia/reperfusion during transplantation is known to lead to very high ROS levels in the transplanted tissue [76,77,78,79]. Given the results shown here, it is tempting to speculate that ischaemia/reperfusion could also result in increased CMV reactivation in the allograft. Indeed, ischaemia/reperfusion has been shown to coincide with increased reactivation frequencies in mouse models of cytomegalovirus [80]. Our results raise the possibility that the burden of CMV reactivation in the transplant setting could be reduced by lowering ROS levels in a block and lock approach. Intriguingly, clinical trials are already underway to reduce ROS levels during transplantation due to the severe damage caused by ROS in transplanted tissue [79]. It would be extremely interesting to test if this has the added benefit of reducing HCMV reactivation.

## 4. Materials and Methods

### 4.1. Cells

Primary CD14+ monocytes were isolated from peripheral blood of healthy donors or from apheresis cones (NHS Blood & Transport Service) as described by Poole et al. [81]. Briefly, peripheral blood mononuclear cells (PBMC) were separated from whole blood by density-gradient centrifugation with Histopaque 1077 (Sigma, St. Louis, MO, USA), and from this, monocytes were isolated by magnetic-activated cell sorting (MACS) with CD14+ microbeads (Miltenyi Biotech, Bergisch Gladbach, Germany). Monocytes were maintained in X-Vivo15 media (Lonza) at 37 °C in a 5% CO_2_ atmosphere.

Monocytes were differentiated into monocyte-derive dendritic cells (moDCs) by the addition of 1000 U/mL IL-4 (Proteintech, Rosemont, IL, USA) and 1000 U/mL GM-CSF (granulocyte-macrophage colony-stimulating factor, Peprotech) for 5 days to generate immature DCs, followed by 2 days stimulation with 50 ng/mL LPS (lipopolysaccharide, Invivogen, Toulouse, France) to produce mature DCs. M2 macrophages were generated by the addition of 20 ng/mL IL-4 and 50 ng/mL M-CSF (macrophage colony-stimulating factor, Proteintech) for 6 days as described in Rostam et al. [82]. 

Cocultures to assess release of infectious virus were performed by addition of 3 × 10^4^ HFFs per well of a 96-well plate. Media was replaced with a mixture of 50% DMEM (supplemented with 10% FBS) and 50% X-Vivo 15. 

### 4.2. Inhibitors

The IKKα inhibitor ΒAΥ11-7082 was obtained from Santa Cruz. The c-Fos inhibitor T5524 was procured from Cayman Chemical. The MEK1+2 inhibitor U0126 was obtained from Source BioScience. The p38 MAPK inhibitor SB203580 was obtained from Sigma. Carbonyl cyanide-p-trifluoromethoxyphenylhydrazone (FCCP, Abcam, Cambridge, UK) was used at a concentration of 20 μM. 

### 4.3. Human Cytomegaloviruses

WT and Δβ2.7 TB40/E-SV40-GFP were kind gifts from Eain Murphy and are described in [17] and [13] respectively. WT and Δβ2.7 Toledo viruses were kind gifts from Gavin Wilkinson and have been described in McSharry et al. [83]. 

Monocytes were infected with virus at a multiplicity of infection (MOI) (based on titration on RPE-1 cells), and incubated for 3 h, before media was washed off twice with PBS and replaced with fresh X-Vivo15 media. 

Human cytomegaloviruses were propagated by inoculation of HFFs at an MOI of 0.1. At 90% infection, supernatant was harvested every 2–3 days and centrifuged for 10 min at 1500× *g* and frozen at −80 °C. To concentrate virus, supernatants were defrosted and spun for 2 h at 14,500× *g* in an Avanti-J25 centrifuge with a JLA-16.250 rotor (Beckman Coulter). The pellet was resuspended in X-Vivo15 media, aliquoted and stored at −80 °C. Before use, concentrated virus was defrosted and spun for 2 min at 800× *g* to further remove any cellular debris.

### 4.4. Fluorescence Microscopy

For intracellular staining, cells were fixed in 1% paraformaldehyde (PFA) for 15 min before being treated with 70% ethanol (EtOH) for 20 min at −20 °C. Cells were then washed once in PBS and blocked in 5% *w*/*v* milk powder dissolved in tris buffered saline (TBS: 50 mM Tris-Cl at pH 7.5, 150 mM NaCl) for 1 h. Monocytes stained for NF-κB were also blocked with Human TruStain FcX™ (Biolegend, San Diego, CA, USA) according to the manufacturer’s instructions. Primary antibody was added at the appropriate dilution (see Table 1) in 5% milk-TBS overnight at 4 °C and subsequently washed off 3 times in TBS. Secondary antibodies (goat anti-mouse Alexa Fluor 594 (ThermoFisher, A11005), goat anti-rabbit Alexa Fluor 594 (ThermoFisher, Waltham, MA, USA, A21207) or goat anti-rabbit Alexa Fluor 488 (ab150077)) were diluted 1/500 in 5% milk-TBS and incubated with Hoechst 33342 nuclear stain (1 μg/mL, Sigma, St. Louis, MO, USA) for 1 h at room temperature, before 3 washes with TBS. Cells were imaged with a widefield Nikon TE200 microscope and all images were processed with ImageJ software. For ROS reactivation experiments, the total number of pp28-positive monocytes were counted in the entire 96 well. 

For surface staining of differentiation markers, cells were fixed in 2% PFA for 10 min before being blocked for 1 h in 5% milk-TBS. Cells were incubated with directly conjugated antibodies at the indicated concentration overnight at 4 °C. Antibodies were washed off 3 times in and cells were imaged on an Olympus IX81 wide-field fluorescence microscope.

Mitochondrial membrane potential was measured using tetramethylrhodamine, ethyl ester (TMRE, Abcam, Cambridge, UK). Monocytes were stained with 50 nM TMRE for 30 min at 37 °C in the dark, before being washed once with PBS. 

Superoxide was stained using MitoSOX™ Red (ThermoFisher, Waltham, MA, USA). MitoSOX reagent was added to monocytes at 1 μM for 10 min at 37 °C in the dark. Cells were subsequently washed three times with Hank’s buffered saline solution (HBSS, Sigma, St. Louis, MO, USA) and Hoechst stained. 

### 4.5. DNA and RNA Extraction, Reverse Transcription and qPCR

At 1 d.p.i, cells to be harvested for RNA or DNA were washed with citrate buffer (40 mM citrate, 10 mM KCl, 135 mM NaCl) for 1 min, followed by 2× PBS wash to remove any extracellular virions.

Genomic DNA was extracted by cell lysis in Solution A (100 mM KCl, 10 mM Tris-HCl pH 8.3, 2.5 mM MgCl_2_) and Solution B (10 mM Tris-HCl pH 8.3, 2.5 mM MgCl_2_, 1% Tween-20, 1% NP-40, 0.4 mg/mL Proteinase K), incubation at 60 °C for 1 h and then incubation at 95 °C for 10 min as described by Roback et al. [84]. 

RNA was harvested with 400 μL Trizol reagent (Ambion) per 10^6^ cells, incubated for 3 min at room temperature, and then stored at −80 °C. RNA was subsequently purified following phenol/chloroform extraction, using the RNeasy Mini Kit (Qiagen, Hilden, Germany) according to the manufacturer’s instructions. Contaminating genomic DNA was selectively degraded with gDNA Wipeout Buffer (Qiagen, Hilden, Germany), and RNA was reverse transcribed using the QuantiTect Reverse Transcription kit (Qiagen, Hilden, Germany) according to the manufacturer’s instructions. 

DNA and cDNA levels were assessed by qPCR, using an ABI 7500 Fast Real Time PCR machine, MicroAmp Fast Optical 96-well reaction plates and Luna^®^ Universal qPCR Master (New England Biotech, Ipswich, MA, USA) according to the manufacturers’ protocol. Primers used are detailed in Table 2. Relative gene expression for all other targets was analysed using ΔΔCt values. 

### 4.6. Flow Cytometry 

Adherent monocytes were first detached from plates using ice-cold PBS supplemented with 2.5 mM EDTA and pipetting up and down. Monocytes were then washed once with PBS and blocked with 1/50 normal mouse serum for 15 min at RT. Cells were stained with the antibodies listed in Table 3 or the equivalent concentration of isotype control antibody for 15 min at RT, washed once in PBS and analysed on a BD Accuri C6 instrument. Live cells were gated by forward and side scatter and final graphs were produced with FlowJo^TM^ software, (FlowJo v9, BD Biosciences, Franklin Lakes, NJ, USA).

### 4.7. Statistics

Statistical calculations (Student’s *t*-test, ANOVA) were performed using GraphPad Prism 9.0 software. ns = not significant, * = *p* < 0.05, ** = *p* < 0.01, *** = *p* < 0.001, **** = *p* < 0.0001.

## 5. Conclusions

In conclusion, we have demonstrated that the HCMV long non-coding RNA, β2.7, is essential for the establishment and maintenance of viral latency in CD14+ monocytes. During infection, β2.7 expression reduces ROS levels and, by doing so, suppresses the activation of NF-κB which, in turn, prevents activation of viral lytic gene expression from the viral MIEP promoter. In support of this, artificially increasing ROS levels in latently infected cells activates NF-κB and induces HCMV reactivation from latency. These findings could provide new targets for preventing HCMV reactivation in clinically important settings, especially given that ROS levels have been shown to be induced in transplanted tissue. 

## Figures and Tables

**Figure 1 ijms-23-11017-f001:**
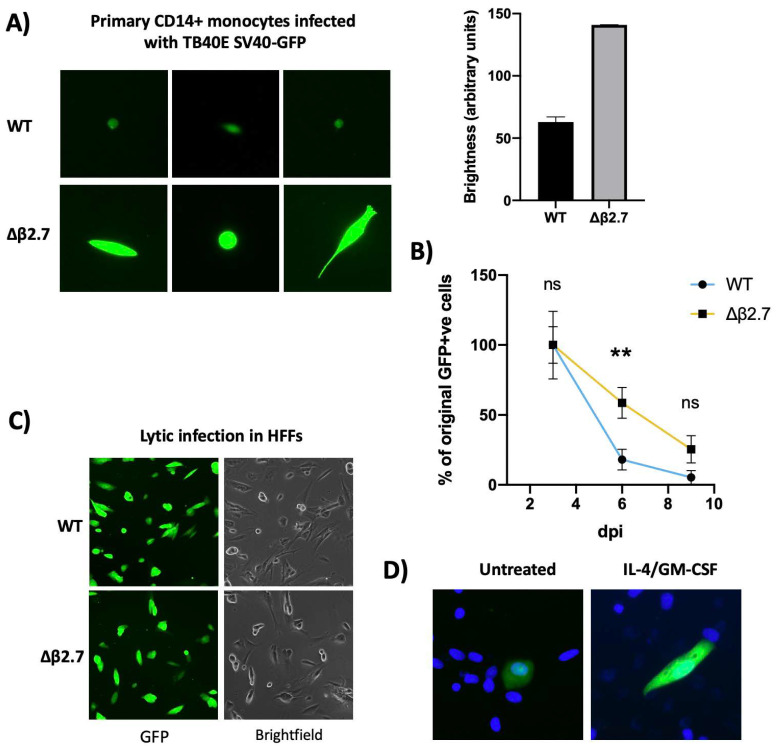
Δβ2.7 virus differs in both intensity and duration of GFP expression in CD14+ cells. (**A**,**B**) Primary CD14+ monocytes were infected with WT SV40-GFP TB40E (WT) or Δβ2.7 SV40-GFP TB40E (Δβ2.7) virus at an MOI of 5. (**A**) Images were captured at 7 d.p.i using a 20× lens and quantified on the right using densitometry analysis from ImageJ software. (**B**) The number of GFP-positive cells in each condition were also counted at 3, 6 and 9 d.p.i and the mean and standard deviation from experiments, performed in triplicate, were plotted. The absolute number of GFP-positive cells at 3 d.p.i was 275.9 ± 86.7 for WT infection and 309.8 ± 87.9 for Δβ2.7 infection. Significant difference was calculated using a 2-way ANOVA with Sidak’s post hoc testing. ** = *p* < 0.01. ns = not significant. (**C**) Human foreskin fibroblasts were infected with WT or Δβ2.7 virus at an MOI of 0.5 and imaged at 3 d.p.i on a 10× lens. (**D**) CD14+ monocytes were infected with WT virus. Half of these cells were differentiated and matured with IL-4/GM-CSF + LPS. Images were taken at 7 d.p.i using the 20× lens on a fluorescence microscope.

**Figure 2 ijms-23-11017-f002:**
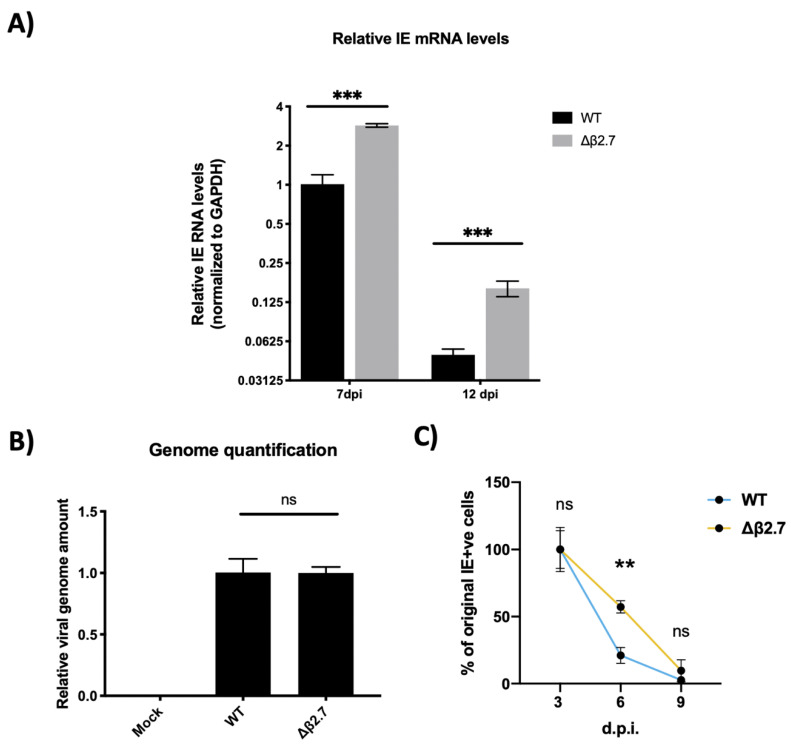
The Δβ2.7 deletion virus fails to silence major IE genes in primary monocytes. CD14+ monocytes were infected with WT SV40-GFP TB40E (WT) or Δβ2.7 SV40-GFP TB40E (Δβ2.7) virus at an MOI of 3. (**A**) RNA was harvested and assayed for levels of IE72/86 and the housekeeping gene GAPDH by RTqPCR at 7 d.p.i and 12 d.p.i. Graphs show mean and standard deviation of three replicates. Significance was determined by two-tailed Student’s *t*-test. *** = *p* < 0.001. (**B**) DNA was harvested from infected monocytes at 1 d.p.i and assessed for levels of the promotor regions of UL44 and GAPDH by qPCR. A significant difference between WT and Δβ2.7 genome levels was determined with a two-tailed Student’s *t*-test. ns = not significant. (**C**) Infected monocytes were fixed at 3, 6 and 9 d.p.i and stained for IE72/86. IE-positive monocytes from 3 replicates were enumerated. Graph shows the mean % and standard deviation of IE-positive cells at 6 and 9 d.p.i relative to 3 d.p.i. Significant difference was determined by a 2-way ANOVA with Sidak’s multiple comparison testing. ** = *p* < 0.01.

**Figure 3 ijms-23-11017-f003:**
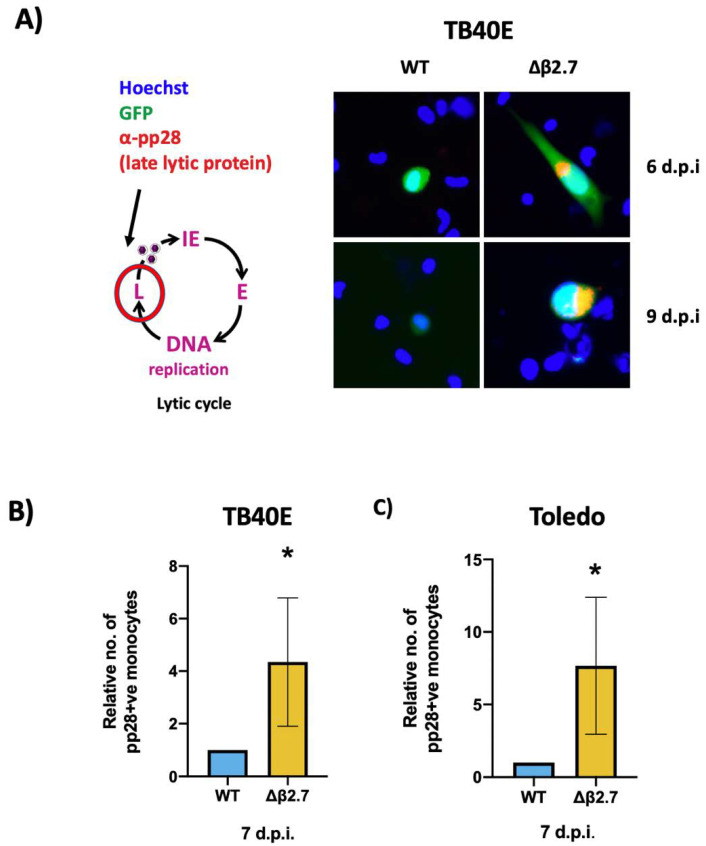
Deletion of β2.7 leads to increased expression of the late lytic protein pp28 in infected monocytes. (**A**) CD14+ monocytes were infected with WT SV40-GFP TB40E (WT) or Δβ2.7 SV40-GFP TB40E (Δβ2.7) virus at an MOI of 5 and fixed and stained for the late lytic protein, pp28, at 6 and 9 d.p.i. Images were captured using a 20× lens. (**B**,**C**) CD14+ monocytes were infected with WT or Δβ2.7 viruses from (**B**) TB40E or (**C**) Toledo strains at an MOI of 5. At 7 d.p.i, cells were fixed and immunostained for pp28. Graph shows the mean number of pp28-positive monocytes relative to WT over 3 independent biological repeats. Error bars show standard deviation. Significance was calculated using a one-tailed Student’s *t*-test. * = *p* < 0.05.

**Figure 4 ijms-23-11017-f004:**
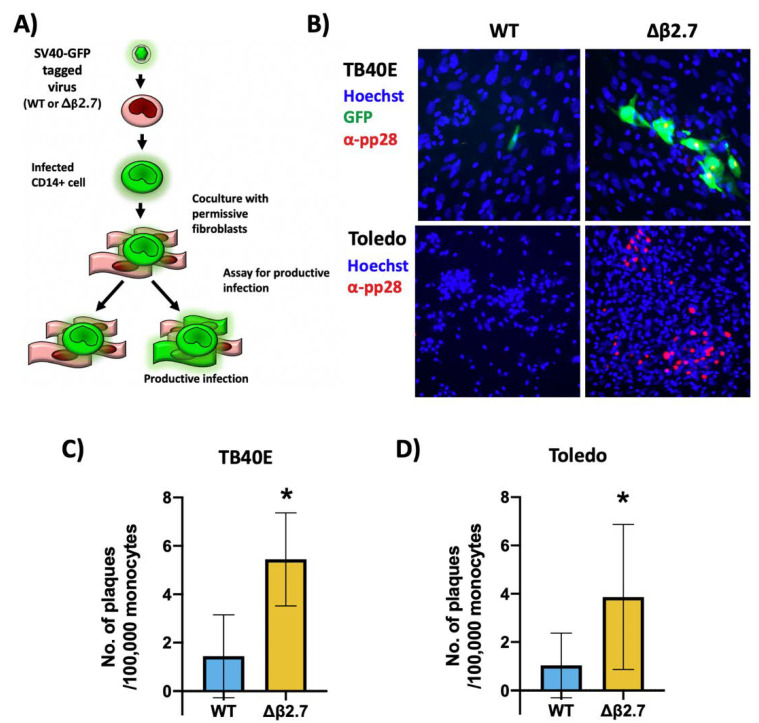
A sub-population of Δβ2.7 infected monocytes undergo a lytic infection in monocytes. (**A**) Diagram of the coculture method to assay for the release of infectious virus. (**B**–**D**) CD14+ monocytes were infected with (**B**,**C**) WT SV40-GFP TB40E (WT) or Δβ2.7 SV40-GFP TB40E (Δβ2.7) virus or (**D**) Toledo strains of WT or Δβ2.7 virus at an MOI of 5. 5 d.p.i, HFFs were co-cultured with the infected monocytes for a further 3 days. Plates were then fixed and stained for pp28, imaged on a 10× lens, (**B**) and the number of plaques counted, (**C**,**D**). Numbers represent the mean of 3 biological repeats for (**C**) and 5 biological repeats for (**D**). * = *p* < 0.05. Significance was determined by Student’s *t*-test.

**Figure 5 ijms-23-11017-f005:**
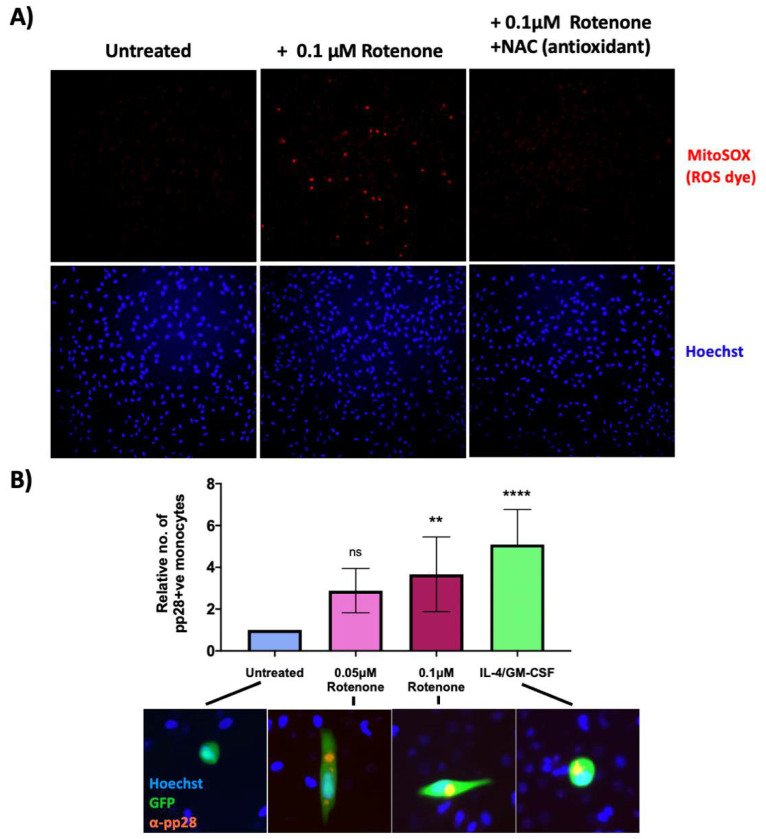
Low levels of rotenone induce superoxide formation in monocytes and prevent the establishment of HCMV latency. (**A**) CD14+ monocytes were treated with 0.1 μM rotenone or 0.1 μM rotenone with 1 mM NAC (antioxidant). 4 h post treatment, cells were stained with the superoxide dye, MitoSOX and visualised by fluorescence microscopy using a 10× lens. (**B**) CD14+ monocytes were infected with WT TB40E SV40-GFP virus at an MOI of 5. At 3 h.p.i, CD14+ cells were left untreated, treated with IL4/GM-CSF + LPS as a positive control for reactivation, or treated with two different concentrations of rotenone (0.05 μM or 0.1 μM). 7 d.p.i, cells were fixed and stained for the late lytic protein, pp28 as a marker of lytic infection. The graph shows the mean relative number of pp28-positive monocytes in each condition over 3 biological repeats. Error bars represent standard deviation. Significant increases relative to ‘untreated’ cells were determined using a nested one-way ANOVA with Dunnett’s post hoc testing. ns = not significant, ** = *p* < 0.01. **** = *p* < 0.0001. Merged colour channel images (20×) for each condition were produced on ImageJ.

**Figure 6 ijms-23-11017-f006:**
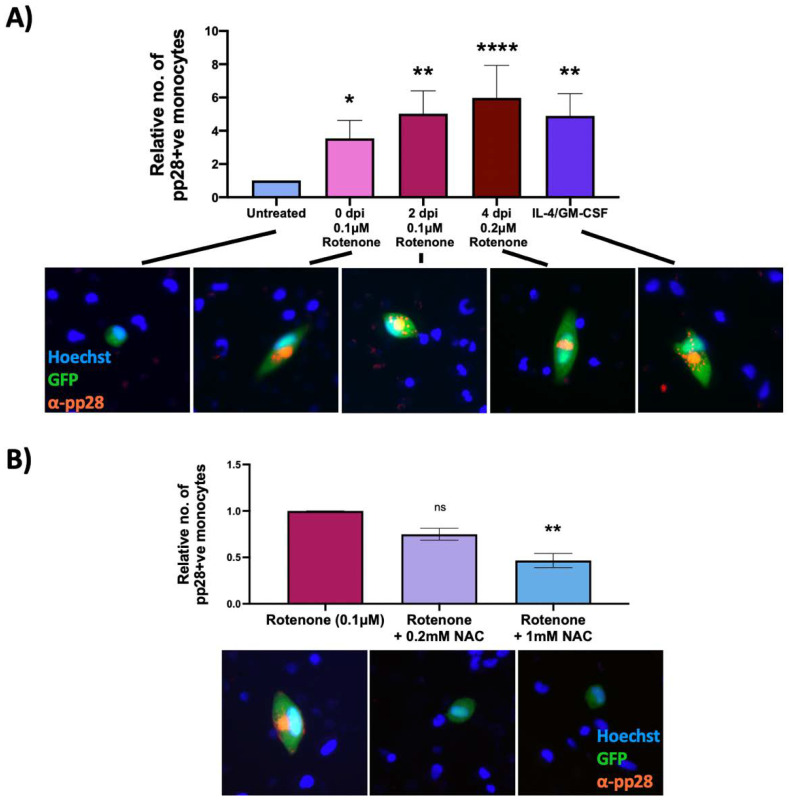
Rotenone induces reactivation of latent HCMV, which is prevented by an antioxidant. CD14+ monocytes were infected with WT TB40E SV40-GFP virus at an MOI of 5. Cells were (**A**) treated with rotenone at 0 (3 h.p.i), 2 or 4 d.p.i, or with IL4/GMCSF + LPS as a positive control for reactivation or (**B**) at 2 d.p.i were treated with 0.1 μM rotenone and increasing concentrations of the antioxidant, NAC. At 7 d.p.i, all cells were stained for the late lytic protein, pp28, and the number of pp28-positive monocytes counted. Graphs show mean relative numbers of pp28-positive monocytes in each condition from at least three independent repeats using 6 × 10^5^ cells/repeat. Error bars represent standard deviation. Significant differences were determined using a nested one-way ANOVA with Dunnett’s post hoc testing. ns = not significant, * = *p* < 0.05, ** = *p* < 0.01, **** = *p* < 0.0001. Merged colour channel images for each condition were produced on ImageJ (20×). Exposure of green channel was reduced in some images for clarity.

**Figure 7 ijms-23-11017-f007:**
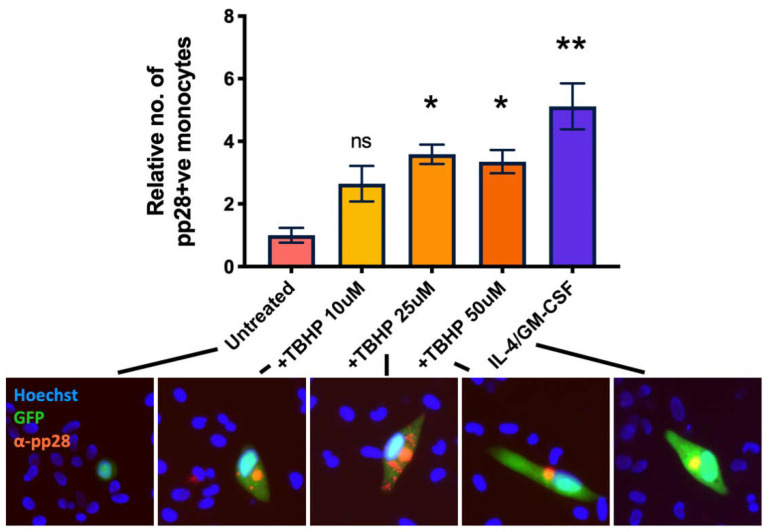
Tert-butyl hydroperoxide also prevents the establishment and maintenance of latency. CD14+ monocytes were infected with WT TB40E SV40-GFP virus at an MOI of 5. 2 d.p.i, cells were treated with increasing amounts of TBHP, or with IL-4/GM-CSF + LPS as a positive control for reactivation. 7 d.p.i, cells were fixed and stained for the late lytic protein, pp28. pp28-positive monocytes were enumerated for each condition. Error bars represent the standard deviation from experiments performed in triplicate. Significant difference from untreated cells was determined by a one-wat ANOVA with Dunnett’s post hoc testing. ns = not significant, * = *p* < 0.05, ** = *p* < 0.01. Merged colour channel images for each condition were produced on ImageJ (20×). Exposure of green channel was reduced in some images for clarity.

**Figure 8 ijms-23-11017-f008:**
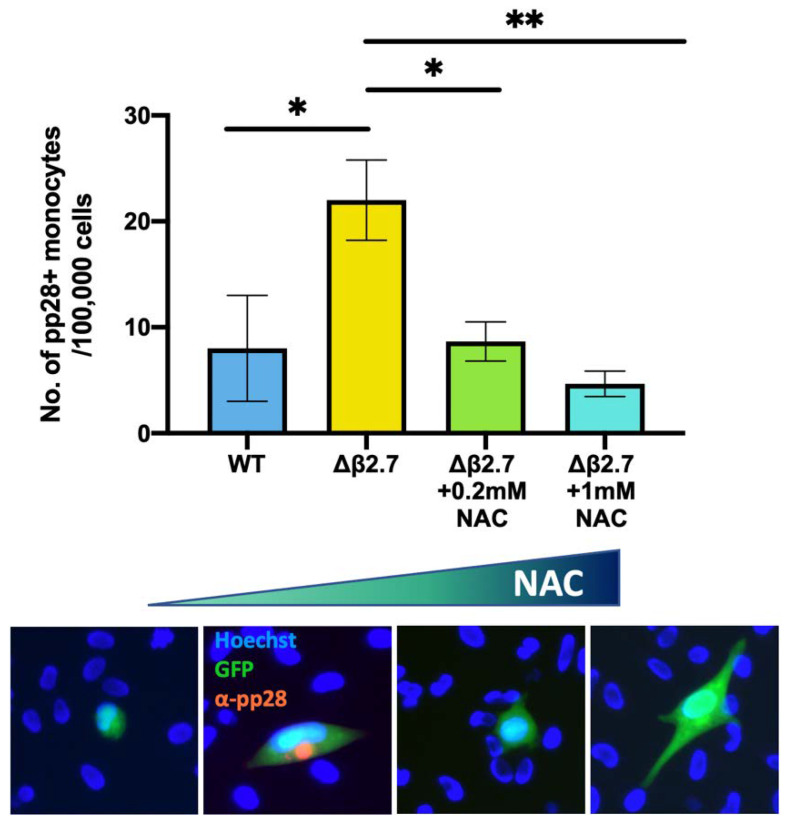
The antioxidant, NAC, reduces the number of Δβ2.7 infected monocytes that fail to establish or maintain latency. CD14+ monocytes were pretreated with the indicated concentrations of the antioxidant, NAC, for 2 h, and were then infected with Δβ2.7 TB40E SV40-GFP (Δβ2.7) virus at an MOI of 5. Every 2 days, cells were provided with fresh media containing the appropriate amount of NAC. At 7 d.p.i, cells were fixed and stained for the late lytic protein, pp28 and pp28-positive monocytes were counted for each condition performed in triplicate. Error bars show standard deviation. Significance was calculated using a one-way ANOVA with Dunnett’s post hoc testing. * = *p* < 0.05, ** = *p* < 0.01. Merged colour channel images for each condition were produced on ImageJ (20×). Exposure of green channel was reduced in some images for clarity.

**Figure 9 ijms-23-11017-f009:**
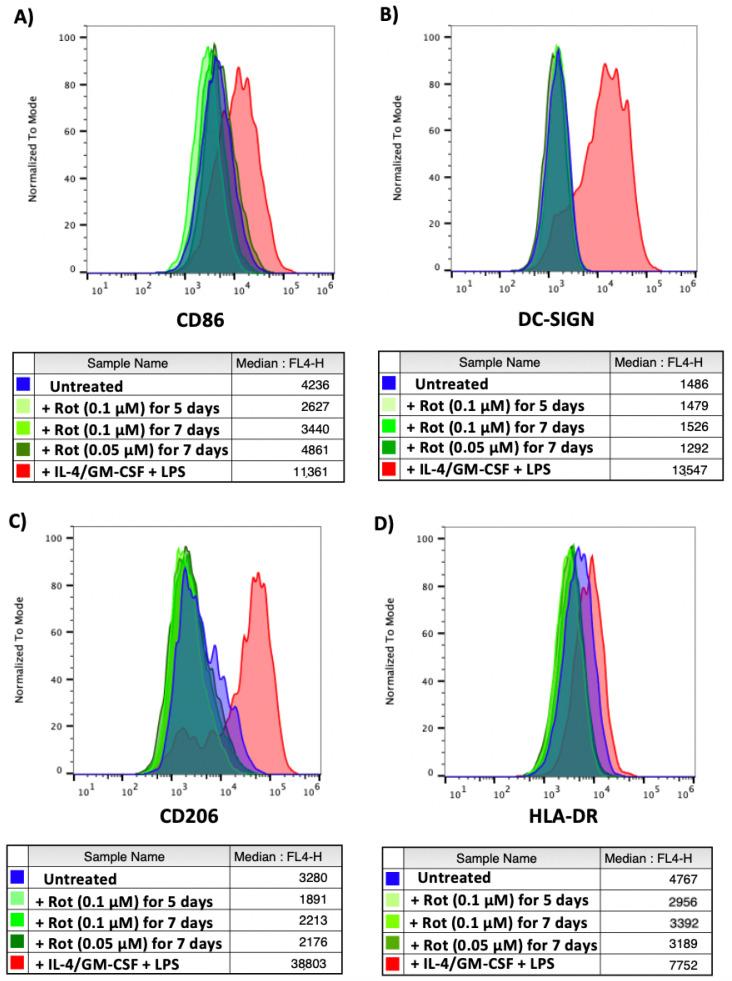
Monocytes treated with rotenone show no signs of differentiation. CD14+ monocytes were left untreated, differentiated and matured with IL-4/GM-CSF + LPS or treated with rotenone for the specified time and concentrations. Monocytes were harvested, stained for (**A**) CD86, (**B**) DC-SIGN, (**C**) CD206 or (**D**) HLA-DR and analysed by flow cytometry. Isotype controls shown in Appendix A.

**Figure 10 ijms-23-11017-f010:**
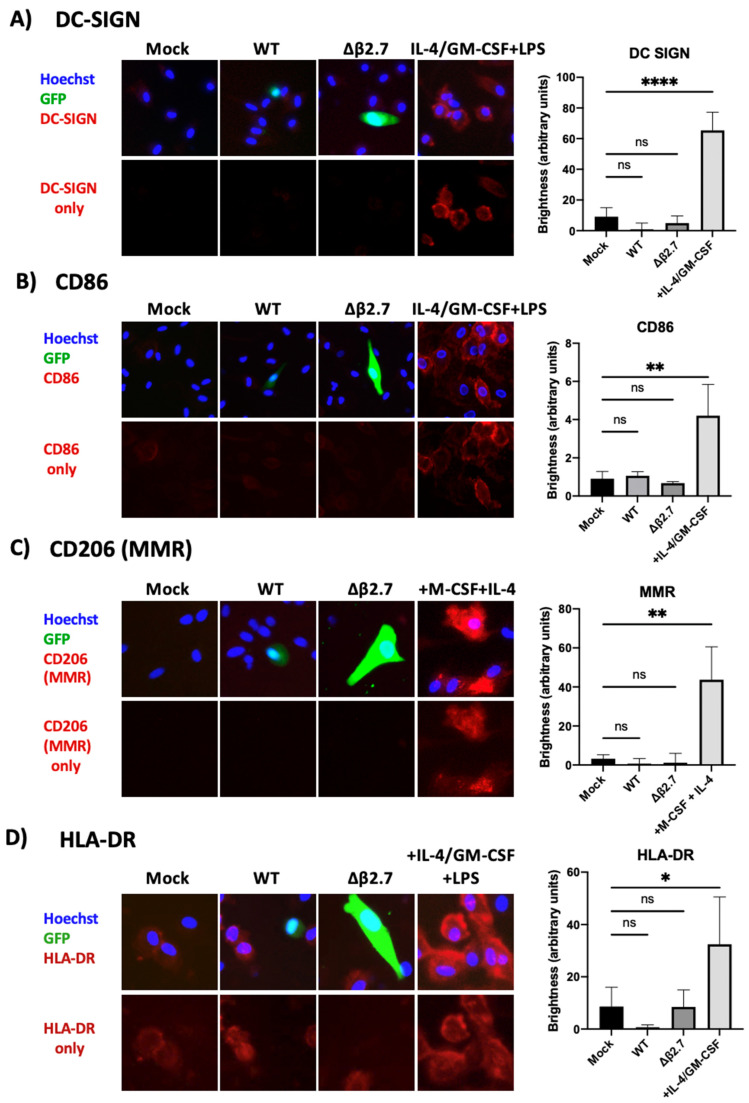
Δβ2.7 virus infected cells show no evidence of differentiation. CD14+ monocytes were mock infected or infected with WT TB40E SV40-GFP (WT) or Δβ2.7 TB40E SV40-GFP (Δβ2.7) at an MOI of 5 and fixed at 7 d.p.i. Additional primary monocytes were also differentiated to: (**A**,**B**,**D**) dendritic cells with IL-4, GM-CSF and LPS, or (**C**) M2 macrophages with M-CSF and IL-4. Fixed cells were stained for (**A**) DC-SIGN, (**B**) CD86, (**C**) CD206 (MMR) and (**D**) HLA-DR and representative photos were taken on a fluorescence microscope using a 20× lens. Brightness was quantified using densitometry analysis on ImageJ software. Significant difference was determined using a one-way ANOVA with Dunnett’s multiple comparison testing. ns = not significant, * = *p* < 0.05, ** = *p* < 0.01, **** = *p* < 0.0001.

**Figure 11 ijms-23-11017-f011:**
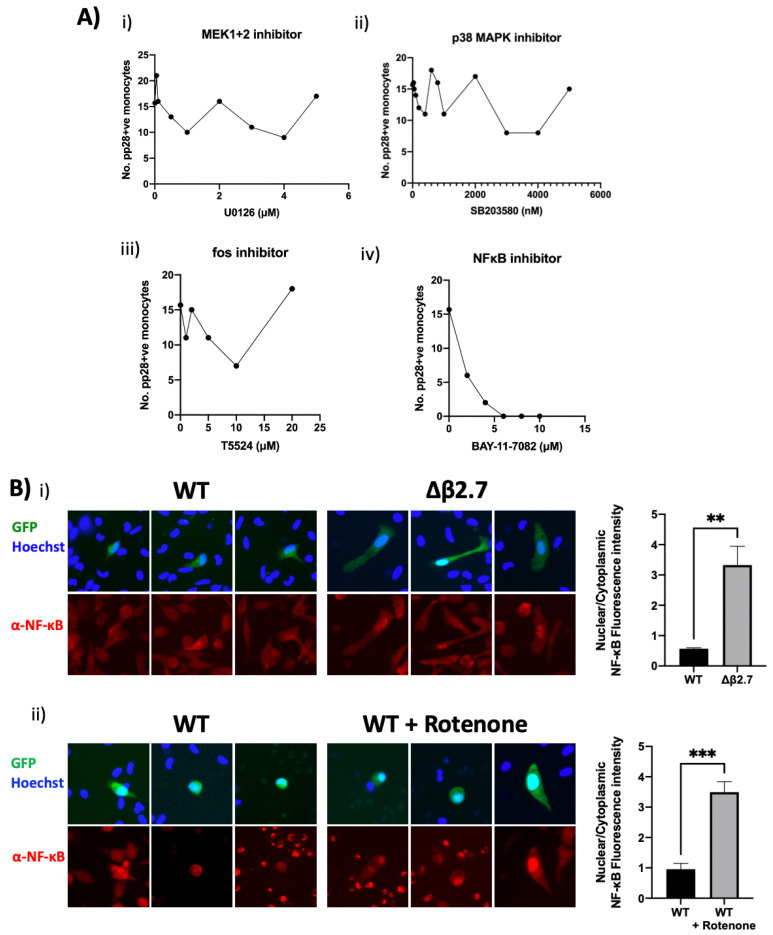
NF-κB activation is essential for ROS-mediated reactivation. (**A**) CD14+ monocytes were infected with Δβ2.7 TB40E SV40-GFP (Δβ2.7) virus at an MOI of 5. At 3 h.p.i, cells were treated with the indicated concentrations of (i) U0126 (MEK1/MEK2 inhibitor), (ii) SB203580 (p38 MAPK inhibitor), (iii) T5524 (fos inhibitor) or (iv) BAY-11-7082 (NF-κB inhibitor). At 7 d.p.i, cells were fixed, stained for pp28, and the number of pp28+ monocytes enumerated. (**B**) CD14+ monocytes were infected with WT TB40E SV40-GFP (WT) or Δβ2.7 TB40E SV40-GFP (Δβ2.7). (i) At 3 d.p.i cells were fixed and stained for NF-κB and imaged under the fluorescence microscope. (ii) As in (**B**(i)), but with rotenone treatment at 4 d.p.i and fixation at 7 d.p.i. Fluorescence intensity was quantified using densitometry analysis in ImageJ and is presented as a ratio of nuclear: cytoplasmic brightness. ** = *p* < 0.01, *** = *p* < 0.001.

**Table 1 ijms-23-11017-t001:** List of antibodies used in immunofluorescence experiments.

Antibody	Company and Catalogue No.	Dilution
anti-IE	Argene, 11-003	1/1000
anti-pp28	Abcam, ab6502	1/1000
anti-NF-κB	Abcam, ab16502	1/500
anti-HLA-DR-APC	Biolegend, #307609	1/200
anti-DC-SIGN-APC	Biolegend, #330107	1/200
anti-CD86-APC	Biolegend, #305412	1/200
anti-MMR-Alexa Fluor 647	Biolegend, #321116	1/200
Mouse IgG2a κ APC	Biolegend, #400219	Equivalent concentration to test antibody
Mouse IgG1 κ APC	Biolegend, #400119	Equivalent concentration to test antibody

**Table 2 ijms-23-11017-t002:** Primers used in qPCR and RTqPCR.

Target	Primer Sequence
IE F	GTC CTG ACA GAA CTC GTC AAA
IE R	TAA AGG CGC CAG TGA ATT TTT CTT C
GAPDH F	TGC ACC ACC AAC TGC TTA GC
GAPDH R	GGC ATG GAC TGT GGT CAT GAG
GAPDH promoter F	CGG CTA CTA GCG GTT TTA CG
GAPDH promoter R	AAG AAG ATG CGG CTG ACT GT
UL44 promoter F	AAC CTG AGC GTG TTT GTG
UL44 promoter R	CGT GCA AGT CTC GAC TAA G

**Table 3 ijms-23-11017-t003:** Flow cytometry antibodies.

Antibody	Source
anti-HLA-DR-APC	Biolegend, #307609
anti-DC-SIGN-APC	Biolegend, #330107
anti-MMR Alexa Fluor 647	Biolegend, #321116
anti-CD86-APC	Biolegend, #305412
Mouse IgG2a κ APC	Biolegend, #400219
Mouse IgG1 κ APC	Biolegend, #400119

## Data Availability

Data is contained within the article or Appendix A.

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
