# Peer review of "The Human Cytomegalovirus β2.7 Long Non-Coding RNA Prevents Induction of Reactive Oxygen Species to Maintain Viral Gene Silencing during Latency"

_ijms, 2022, doi:10.3390/ijms231911017_

Round 1

Reviewer 1 Report

The manuscript titled “The human cytomegalovirus β2.7 long non-coding RNA prevents induction of reactive oxygen species to maintain viral gene silencing during latency” found that higher level of ROS reactivated HCMV replication from latency in monocytes, which depended on NF-kB activation, and β2.7 contributed the establishment of latency through reduction of ROS. These findings benefit to understand the mechanism of HCMV reactivation in transplanted patients, and are of particular significance in further clinical application. There are some minor points needed to be explained. 

1.     It is interesting about the percentages of GFP positive cells to the cells infected by Δβ2.7 SV40-GFP TB40E at longer time points than 9 dpi in Fig 1B. Does the Δβ2.7 fail to silence the GFP expression over time?

2.     In figure 1B and 2D, there are significant differences between WT and Δβ2.7 mutant in viral replication and IE transcription at 6 dpi, while the difference was not so obvious at 9 dpi. Could you explain the phenomenon?

3.     The authors showed the increase of IE transcript at 7 and 12 dpi in figure 2A and 2B respectively, and a non significant difference of viral genome between the two samples at 1 dpi in figure 2C. I think it would be better to supplement results of the viral genome quantification at 7 and 12 dpi, too.

4.     In figure 6A, to reactivate the replication of HCMV, why the rotenone treatment was started at 0, 2 and 4 dpi instead of 6 dpi when there was a significant decrease of viral replication in Figure 1B? And why the work concentration of rotenone was increased to 0.2μM at 4 dpi?

5.     Although the authors have provied enough evidences to their conclusion, an additional comparison of the abilities for estabolishing latency and reactivating from latency betwen WT and Δβ2.7 mutant in condition of retenone-induced high ROS will certainly be helpful.

6.     Please check the following descriptions:

(1)     Line 93-94, ‘……in monocytes infected with Δβ2.7 virus and infected CD14+ monocytes with either the TB40E SV40-GFP WT or Δβ2.7 viruses’.

(2)     Line 107, ‘A significant difference between WT and Δβ2.7 genome levels was determined with a two tailed student’s t-test. ns (not significant) = p >0.05.’

(3)     Line 130, ‘It is known that that IE expression……’

Reviewer 2 Report

In this study, Perera and Sinclair investigated the role of the HCMV β2.7 lncRNA on latent vs lytic infection of monocytes. They found that HCMVΔβ2.7 is more likely to enter the lytic cycle than WT HCMV, suggesting that the β2.7 lncRNA contributes to the establishment or maintenance of latency in monocytes. As β2.7 is known to inhibit ROS formation, the authors speculated that increased ROS levels in monocytes might promote HCMV lytic replication. Indeed, they could show that rotenone, a ROS inducer, promotes the lytic cycle, while NAC, a ROS neutralizer, reduces it. They further show that rotenone or HCMVΔβ2.7 infection do not induce monocyte differentiation, but rather promote NF-κB activation, which is responsible for the increase in lytic cycle activation.

Overall, this is a very nice paper describing interesting new findings on the role of the HCMV β2.7 lncRNA in the establishment or maintenance of latency. The effects of β2.7 deletion are not overwhelming but rather modest (approx. 5-fold increase in lytic replication compared to WT), but they are significant in all experiments. My only major concern is that the authors tend to oversell their data. This should be corrected.

Specific comments

1.  On several occasions throughout the manuscript, the authors state that the absence of β2.7 results in a lytic rather than latent infection in monocytes (e.g., lines 61, 163, 205). This statement is not consistent with the data presented. Fig 4CD and Fig 8 show that the percentage of infected monocytes expressing pp28 and releasing infectious virus is very low in cells infected with HCMVΔβ2.7 – higher than for WT HCMV-infected cells but still low.

2.  The authors claim that “β2.7 is essential for establishing and maintaining HCMV latency” (Abstract, line 15) and that “HCMV lacking the viral β2.7 gene fails to establish latency” (line 367). There is no data in the manuscript supporting these claims. Only a small percentage of HCMVΔβ2.7-infected cells are lytically infected. Conversely, one has to assume that the majority of the Δβ2.7-infected cells are latently infected, unless the authors have data proving that this is not the case. If such data exists, it must be included, otherwise the statements have to be adjusted.

3.  Fig 3.  Did the authors check for E and L gene expression by Western blot? Or is the number of lytically infected cells too low?

4.  Several signaling pathways are known to be involved in the reactivation of latent HCMV, not just the ROS - NF-κB pathway. It would be helpful if the authors could use the Discussion to put their work in context.

Minor points

Fig 1. WT and Δβ2.7-infected cells are shown horizontally in panel A and vertically in panel C. It would be nice to keep the arrangements consistent.

Line 88. It seems more likely that increased GFP expression indicates genome amplification.

Spelling. NF-κB, not NFκB; Hoechst, not Hoescht (Fig 11); space between numbers and units are frequently missing, e.g. 0.1µM; line 180, actively respiring mitochondria (not mitochondrial).

Reviewer 3 Report

This manuscript presents data supporting that hypothesis that the HCMV lncRNA b2.7 plays an important role in repressing IE gene expression during the establishment of latency in human monocytes. The authors propose that b2.7 achieves this through inhibiting the accumulation of reactive oxygen species and show that several ROS inducers can prevent latency establishment. The study concludes with links between ROS signaling and NFkB activation. The function of the b2.7 lncRNA has long been a mystery, especially given its abundance during all stages of HCMV infection. While the findings of this study are intriguing, in several instances the experiments lack rigor and the manuscript requires additional experiments and controls to support the author’s conclusions. Significantly lacking were experiments showing that the various infections and/or drug treatments led to an increase in virus production (with the exception of Figure 4) in order to support the conclusions regarding the role of b2.7 and ROS in latency establishment. Additionally, an incomplete reference list made it impossible to verify some of the claims made throughout the manuscript.

Major comments:

By the way the data is presented, it is unclear what proportion of the total infected cell population is GFP positive at the timepoints shown in Figure 1 and how many cells of the total population are capable of latency establishment and reactivation. Figure 1B shows that cultures of both WT and mutant virus have decreasing numbers of GFP positive cells over time – does this reflect cells undergoing viral gene silencing over time with genomes that will be capable of reactivation when appropriately stimulated? This becomes important to know in order to distinguish between viral mutants that are truly incapable of establishing a latent infection and those that are more prone to lytic replication (but a proportion of virus still establishes latency). Is there statistical differences in the number of GFP positive cells between WT and mutant virus in Figure 1B? Is there a difference in the absolute number of GFP positive cells between the WT and mutant virus at the earliest timepoint? This information is critical to understanding the significance of the remainder of the experiments.

Figure 2A and 2B show relative levels of IE transcripts at 7 and 12 dpi. It would be interesting to know how the absolute IE levels change between 7 and 12 days for both the WT and mutant virus. Presumably the IE levels would stay low and/or similar between the 2 timepoints for WT infected cells but how do IE levels change with the mutants? Was there a significant difference in the number of IE positive cells shown in Figure 2D at any timepoint? Similar to Figure 1, it is hard to get a conceptual idea of how many cells in the total population are expressing IEs based on the way the data is presented. Are the absolute number of IE positive cells different between WT and the mutant at the earliest timepoint?

The timepoint chosen for Figure 4 – 5dpi - correlates approximately with the time where the greatest difference in GFP and IE protein is observed between WT and mutant virus. Do the number of plaques produced decline the longer after infection this experiment is performed? Again, this gets to the question of a persistent lytic infection versus a transient increased level of viral replication mediated by b2.7.

Figure 6B is lacking important controls including untreated cells and cells treated only with NAC.

 Figure 8 is lacking controls including WT virus infection and WT virus treated with NAC. Does pre-treatment with NAC affect the ability of the virus to enter the cell and/or deliver the genome to the nucleus?

The data examining differentiation marker expression in infected cells presented in Figure 10 must be quantitated in some manner. Could this experiment be performed by flow cytometry (as in Figure 9)? Since Figure 4 shows that mutant virus-infected cells produce virus when co-cultured on fibroblasts, is it surprising that there is no expression of differentiation markers in these cells or instead is the interpretation that the cells are lytically infected and producing virus without differentiation?

The IF images shown in Figure 11B are not convincing and they show only one infected cell per field of view. In panel ii it appears that WT infection does induce NFkB nuclear translocation, or at best it is unclear. This must be quantitated in some manner. Can the authors speculate why rotenone only induced NFkB translocation in infected cells? Missing is the Mock + rotenone control.

Minor comments:

In Figure 1, the authors refer to higher levels of GFP expression in the mutant infected monocytes – can flow cytometry be used to better support their conclusions?

For Figure 2, the text refers to harvesting DNA at 16hpi, the legend of Figure 2C says 1dpi and text later says 3hpi.

Figure 3 – Do the authors have any idea why the Toledo version of the mutant virus results in more pp28 positive monocytes?

It is unclear how many times the experiment shown in Figure 9 was replicated.

How many times was the experiment in Figure 11A replicated?

Round 2

Reviewer 3 Report

I thank the authors for taking my comments under consideration and making important improvements to the manuscript. I think the inclusion of the sentence regarding the potential death of b2.7-infected cells that occurs at later times of infection is a very important one, as the interpretation of the results in Figures 1B and 2C were difficult to understand with the lack of significance at 9dpi. Finally, the findings that rotenone treatment or infection with the b2.7 mutant does not induce differentiation (at least by the examination of a handful of markers) is very intriguing and suggests that expression of the full complement of viral genes and production of new virus has no effect on the differentiation status of the cell. This is in contrast to other published work (PMC7411144 and references therein) and worthy of comment in the manuscript and more specific study in the future.